Inhibitory effects of Trichoderma asperellum culture filtrates on pathogenic bacteria, Burkholderia pseudomallei

Roopkhan Naritsara 1
Chaianunporn Thotsapol 1
Chareonsudjai Sorujsiri 2
Chaianunporn Kanokporn kanokporn.s@msu.ac.th 3
1 Department of Environmental Science, Faculty of Science, Khon Kaen University , Khon Kaen , Thailand
2 Department of Microbiology, Faculty of Medicine, Khon Kaen University , Khon Kaen , Thailand
3 Faculty of Medicine, Mahasarakham University , Maha Sarakham , Thailand
Tharmalingam Nagendran
Electronic publication date: 2025 Feb 28
Publication date: 2025
Volume: 13
Electronic Location ID: e19051
Received 2024 Oct 7; Accepted 2025 Feb 4
Copyright: ©2025 Roopkhan et al.
Copyright year: 2025
Copyright holder: Roopkhan et al.
License: This is an open access article distributed under the terms of the Creative Commons Attribution License, which permits unrestricted use, distribution, reproduction and adaptation in any medium and for any purpose provided that it is properly attributed. For attribution, the original author(s), title, publication source (PeerJ) and either DOI or URL of the article must be cited.
License URL: https://creativecommons.org/licenses/by/4.0/

Keywords: Trichoderma asperellum, Culture filtrate, Burkholderia pseudomallei, Antibacterials, Anti-biofilm

Funding: Faculty of Science, Khon Kaen University SCG-2022-2 Faculty of Medicine, Mahasarakham University This research was funded by “The research capability enhancement program through graduate student scholarship” Faculty of Science, Khon Kaen University to Thotsapol Chaianunporn and Naritsara Roopkhan with grant number SCG-2022-2 and a research grant from Faculty of Medicine, Mahasarakham University to Kanokporn Chaianunporn. The funders had no role in study design, data collection and analysis, decision to publish, or preparation of the manuscript.

==============================
Background

Burkholderia pseudomallei is a soil- and water-dwelling bacterium that causes the life-threatening infection melioidosis. Patients typically acquire this infection through environmental exposure, so reducing B. pseudomallei levels in the environment could mitigate the risk of infection. Trichoderma asperellum is a biological control agent that synthesizes a diverse range of antimicrobial substances targeting other microorganisms. This study therefore examined the antibacterial and anti-biofilm activities of T. asperellum culture filtrate against B. pseudomallei.

Methods

The antibacterial activities of T. asperellum culture filtrates, collected at various time intervals, were assessed against B. pseudomallei using the agar well diffusion method. Subsequently, the minimum inhibitory concentrations (MICs), minimum bactericidal concentrations (MBCs), and anti-biofilm activities of the culture filtrate exhibiting the highest inhibitory effect were determined. Bactericidal efficacy was further evaluated via a time-kill assay. The mechanisms underlying inhibition were then investigated using scanning electron microscopy and crystal violet uptake assays.

Results

Filtrate collected from 7-day old cultures of T. asperellum (TD7) exhibited the strongest inhibitory effect on B. pseudomallei, with an inhibition zone of 30.33 ± 0.19 mm. The MIC of TD7 against B. pseudomallei was 7.81 ± 0.00 mg/mL and the MBC ranged from 7.81 ± 0.00 to 11.72 ± 1.75 mg/mL. Time-kill studies with TD7 confirmed its bactericidal activity, with complete elimination of B. pseudomallei occurring within 30 min treatment at 62.48 mg/mL (8xMIC) and 24 h treatment at 7.81 mg/mL (1xMIC). At a concentration of  7.81 mg/mL, TD7 also significantly reduced B. pseudomallei biofilm formation. Scanning electron microscopy revealed surface roughening and cell shrinkage of TD7-treated B. pseudomallei. TD7-treated bacteria were also found to absorb more crystal violet dye than untreated cells, indicating that TD7 might inhibit and kill B. pseudomallei by disrupting cell membrane permeability.

Conclusions

Our findings demonstrate that T. asperellum culture filtrates possess bactericidal activity and effectively disrupt biofilm formation by B. pseudomallei. This suggests that T. asperellum could potentially be used to reduce the presence of B. pseudomallei in the environment and, consequently, lower the incidence of melioidosis.

Introduction

Burkholderia pseudomallei is a Gram-negative bacterium that causes melioidosis, a serious human and animal infection that can manifest as bacteremia, pneumonia, hepatosplenic abscesses, septic arthritis, and skin or soft tissue infections (Cheng & Currie, 2005; Currie, Dance & Cheng, 2008; Limmathurotsakul et al., 2012; Currie, 2015). This pathogen is widely distributed in soil and water and is endemic in Southeast Asia and Northern Australia (Cheng & Currie, 2005; Currie, Dance & Cheng, 2008; Wuthiekanun et al., 2009; Limmathurotsakul et al., 2016; Hinjoy et al., 2018; Gee et al., 2022; Currie, Meumann & Kaestli, 2023). The majority of melioidosis patients are rice farmers who have been exposed to B. pseudomallei present in wet soil (Suputtamongkol et al., 1999; Limmathurotsakul et al., 2010; Rattanavong et al., 2011; Limmathurotsakul et al., 2013). Reducing B. pseudomallei levels in soil and water may therefore mitigate the risk of exposure and infection.

Several studies have proposed that modification of the physiochemical or biological properties of soil could reduce B. pseudomallei in the environment (Ngamsang et al., 2015; Boottanun et al., 2017; Sermswan & Wongratanacheewin, 2017). B. pseudomallei is known to form biofilms though (Vorachit, Lam & Costerton, 1995), an important survival strategy for adaptation to harsh environmental conditions (Davey & O’toole, 2000; Hall-Stoodley, Costerton & Stoodley, 2004; Kamjumphol et al., 2013). Biofilm formation enables bacteria, including B. pseudomallei, to survive in the presence of chemical agents such as herbicides, pesticides, and disinfectants (Inglis & Sagripanti, 2006; Lima et al., 2020; Freitas et al., 2021) and against host immune responses as well as antibiotic treatments (Sawasdidoln et al., 2010; Duangurai, Indrawattana & Pumirat, 2018; Khamwong et al., 2022). Thus, eliminating bacteria within biofilms is challenging. Two methods have been proposed for eradicating biofilm-forming bacteria in the environment—the first disrupts the biofilm matrix to enhance chemical penetration and the second transitions the bacteria to the more susceptible planktonic form (Davies, 2003; Pakkulnan et al., 2019; Jalil & Ibrahim, 2021; Pakkulnan, Thonglao & Chareonsudjai, 2023).

Trichoderma is a genus of filamentous fungi belonging to the family Hypocreaceae, commonly found in soil and decomposing organic matter. Trichoderma species, including T. asperellum, produce primary and secondary metabolites that promote plant growth (Wu et al., 2017; Herrera-Téllez et al., 2019), inhibit bacterial pathogens of humans (Phupiewkham et al., 2015; Santos et al., 2018; Zhang et al., 2021) and demonstrate antagonistic activity against phytopathogenic fungi (Vizcaino et al., 2005; Vinale et al., 2014; Zeilinger et al., 2016; Herrera-Téllez et al., 2019; Khan et al., 2020). Given that T. asperellum is already mass-produced and distributed to farmers in Thailand for controlling phytopathogenic fungi (Kamkha, 2019; Chamswarng, 2020; Unartngam et al., 2020), it could potentially be repurposed for B. pseudomallei control if it is found to have sufficient activity against this species. To our knowledge, information regarding the effects of Trichoderma on B. pseudomallei is very limited. In this study, we investigated the bactericidal and anti-biofilm properties of T. asperellum culture filtrates against B. pseudomallei. We also investigated the mechanism of action of T. asperellum culture filtrate using scanning electron microscopy to examine cell morphology and a crystal violet uptake assay to assess membrane permeability. These findings provide insights into the potential application of T. asperellum or its culture filtrates as biological or biochemical control agents against soil- and water-dwelling B. pseudomallei.

Materials & Methods

Fungal and bacterial strains

The Trichoderma asperellum strain used in this study was obtained from Khon Kaen Agricultural Technology Promotion Center (Plant Protection) in Khon Kaen Province, Thailand. The two bacterial strains, Burkholderia pseudomallei K96243 and B. pseudomallei H777 (Taweechaisupapong et al., 2005; Kunyanee et al., 2016; Khamwong et al., 2022), were provided by the Melioidosis Research Center at Khon Kaen University Faculty of Medicine.

Fungal cultivation and culture filtrate preparation

T. asperellum was cultured on potato dextrose agar (PDA) at 30 °C for 5 days. Thereafter, mycelial discs were cut and placed into 50 mL of potato dextrose broth (PDB), incubated at 30 °C in an incubator shaker at 125 rpm. The culture supernatants were collected on days 3, 5, 7, 9, 11 and 14 by centrifugation at 4,000 rpm for 10 min at 4 °C and filtered through a 0.2 µm filter before freeze-drying. The dried culture filtrates were weighed and dissolved in sterile distilled water to achieve a final concentration of one g/mL and then stored at 4 °C as stock solutions for further use (Sangdee et al., 2016; Chaianunporn et al., 2018).

Bacterial culture conditions

B. pseudomallei K96243 and H777 strains, obtained from glycerol stocks, were cultured on Ashdown’s agar at 37 °C for 48 h (Kunyanee et al., 2016; Pakkulnan et al., 2019; Khamwong et al., 2022; Pakkulnan, Thonglao & Chareonsudjai, 2023). A single colony of B. pseudomallei was inoculated into three mL of tryptic soy broth (TSB), Mueller-Hinton broth (MHB) or Luria-Bertani (LB) broth and incubated at 37 °C at 200 rpm for 18 h, until the culture reached a concentration equivalent to a 0.5 McFarland turbidity standard (OD600 = 0.08−0.1). Thereafter, the bacterial cultures were diluted to approximately 106–108 colony forming units (CFU)/mL for the subsequent experiments (Sangdee et al., 2016; Pakkulnan et al., 2019; Chaianunporn, Chaianunporn & Chareonsudjai, 2020; Pakkulnan, Thonglao & Chareonsudjai, 2023).

Preliminary screening of T. asperellum culture filtrates for antibacterial activity by the agar well diffusion method

The antibacterial activity of T. asperellum culture filtrates were examined against B. pseudomallei K96243 using the agar well diffusion method (Sangdee et al., 2016; Chaianunporn et al., 2018). The B. pseudomallei K96243 inoculum in TSB (106 CFU/mL) was spread on Nutrient agar (NA) plates before punching them with a sterile cork borer. One hundred microliters (µL) of T. asperellum culture filtrates from days 3, 5, 7, 9, 11 and 14 at a concentration of 400 mg/mL, along with sterile distilled water (negative control) and 30 µg/mL ceftazidime (CAZ –positive control), were added to each well and incubated at 37 °C for 18 h. The diameters of the zones of inhibition around the culture filtrates and controls were then measured. Six replicates of each filtrate were tested. The culture filtrates with the largest inhibition zone were used for subsequent experiments.

Determination of minimum inhibitory concentrations (MICs) and minimum bactericidal concentrations (MBCs) of T. asperellum culture filtrate against B. pseudomallei

Two-fold serially diluted T. asperellum culture filtrate with concentrations ranging from 500 to 0.24 mg/mL were prepared in 96-well microtiter plates. Subsequently, an inoculum of B. pseudomallei K96243 and H777 in MHB (106 CFU/mL) was added to each well and incubated at 37 °C for 18 h. Growth control wells were included on each plate. The MIC values were documented as the lowest concentration at which no visible bacterial growth was observed. The MBC values were determined by transferring suspension from wells that showed no visible growth onto NA plates. These plates were then incubated at 37 °C for 24 h (Chaianunporn, Chaianunporn & Chareonsudjai, 2020). Six replicates were performed for each B. pseudomallei isolate.

Evaluation of bactericidal activity of T. asperellum culture filtrate by time-kill assay

B. pseudomallei K96243 suspension (106 CFU/mL) was inoculated into TSB tubes containing different concentrations of T. asperellum culture filtrate at 7.81 mg/mL (1 ×MIC), 15.62 mg/mL (2 ×MIC), 31.24 mg/mL (4 ×MIC), and 62.48 mg/mL (8 ×MIC) along with a growth control tube without the culture filtrate were compared in this study. These tubes were incubated at 37 °C. Bacterial enumeration was performed at specified time intervals (0, 2, 4, and 24 h) using the drop plate method (Herigstad, Hamilton & Heersink, 2001). The number of bacteria remaining in each sample was plotted over time to determine the rate of killing. A three log10 reduction in bacterial counts was used as the criterion for bactericidal activity (Sangdee et al., 2016; Thammawat, Sangdee & Sangdee, 2017). Each concentration was tested in triplicate.

Anti-biofilm activity of T. asperellum culture filtrate against B. pseudomallei biofilm

Biofilm formation of B. pseudomallei strain H777 was assessed using crystal violet staining in 96-well microtiter plates (Pakkulnan et al., 2019; Allkja et al., 2020; Pakkulnan, Thonglao & Chareonsudjai, 2023). Two hundred microliters (µL) of B. pseudomallei-inoculated Luria-Bertani (LB) broth (107 or 108 CFU/mL) was dispensed into each well of a 96-well flat-bottomed polystyrene plate. T. asperellum culture filtrate at 3.91 mg/mL (0.5 ×MIC) and 7.81 mg/mL (1 ×MIC) was then added to the cell suspension plates either at 0 h (adhesion stage treatment) or at 24 h (biofilm formation stage treatment). Each treatment plate was further incubated at 37 °C under static conditions. After incubation for 48 h, biofilms were carefully washed three times with 200 µL per well of sterile distilled water. The biofilm in each well was fixed with 99% methanol for 15 min and air dried. Next, the biofilms were stained with 2% w/v crystal violet (the working solution was prepared at a ratio of Solution A (20 g of 85% crystal violet dye dissolved in 100 mL of 95% (v/v) ethanol), Solution B (one g of ammonium oxalate dissolved in 100 mL of distilled water), and distilled water as 1:8:1) for 5 min. Excess stain was gently washed with running tap water until clear. After air-drying, adherent crystal violet stain was dissolved in 200 µL of 33% (v/v) glacial acetic acid. Finally, the solutions were transferred to a new 96-well plate and subsequently diluted five-fold with additional acetic acid. The optical density of each sample at 620 nm (OD620) was measured using a microplate reader (TECAN Safire, Port Melbourne, Australia). The values were calculated by multiplying by 5 to determine the original optical density. Each concentration was tested eight times across three independent experiments.

Use of scanning electron microscopy to detect T. asperellum culture filtrate-induced changes in B. pseudomallei cell morphology

T. asperellum culture filtrate-treated B. pseudomallei K96243 were observed for morphological alterations using a scanning electron microscope. B. pseudomallei cells were suspended in TSB (106 CFU/mL) and treated with 3.91 mg/mL (0.5 ×MIC) and 7.81 mg/mL (1 ×MIC) of T. asperellum culture filtrate at 37 °C for 16 h. The treated samples were then fixed with 2.5% glutaraldehyde at 4 °C for 12 h, before being washed three times with phosphate-buffered saline (PBS). Bacterial cell dehydration was achieved using increasing concentrations of ethanol (30%, 50%, 80% and 100%). After this, the dried samples were mounted onto stubs, coated with 40–60 nm of gold, and observed using a desktop scanning electron microscope (MiniSEM; SEC model SNE-4500M, South Korea) (Chaianunporn et al., 2018).

Detection of B. pseudomallei membrane damage by crystal violet uptake assay

Alterations in the membrane permeability of T. asperellum culture filtrate-treated B. pseudomallei were assessed using the crystal violet uptake assay (Devi et al., 2010; Khan et al., 2017). Suspensions of B. pseudomallei K96243 in TSB (106 CFU/mL) were harvested by centrifugation at 10,000 rpm for 5 min, then washed twice and resuspended in PBS. The cell suspensions were treated with 3.91 mg/mL (0.5 ×MIC) and 7.81 mg/mL (1 ×MIC) of the T. asperellum culture filtrate at 37 °C for 30 min. Cells treated with EDTA (0.25 M) were used as a positive control, and untreated cells were used as a negative control. After that, the bacterial cells were centrifuged at 10,000 rpm for 5 min and then resuspended in PBS containing 10 µg/mL of crystal violet. The cell suspensions were then incubated at 37 °C for 10 min and centrifuged at 10,000 rpm for 5 min. Subsequently, the optical density of the supernatants at 590 nm (OD590) was measured using a UV–VIS spectrophotometer (Naondrop2000C; Thermo Fisher Scientific, Waltham, MA, USA). Each concentration and control were tested in at least four replicates. The absorbance value of the initial crystal violet solution used in the assay was considered as 100%. The percentage of crystal violet uptake was calculated using the following formula: %Crystal violet uptake=OD590of the sample/OD590of crystal violet solution×100.

Statistical analysis

Statistical analyses were performed using one-way ANOVA followed by the Tukey–Kramer test at a significance level of 0.05 for comparing differences between mean values by using R version 4.4.0 (R Core Team, 2024).

Results

Antibacterial activity of T. asperellum culture filtrates against B. pseudomallei by the agar well diffusion method

The T. asperellum culture filtrates collected on days 3, 5, 7, 9, 11 and 14 could all inhibit B. pseudomallei K96243, though the inhibition zone sizes varied significantly among the collection times (one-way ANOVA: F5,30 = 26.58, p = 3.53e−10). Notably, the culture filtrate collected on day 7 (TD7) produced the largest zone of inhibition (Table 1). TD7 was therefore used for determining minimum inhibitory concentration (MIC) and minimum bactericidal concentration (MBC) values in subsequent experiments.

Table 1 Antibacterial activity of T. asperellum culture filtrates collected at different time points against B. pseudomallei K96243 assessed by zone of inhibition measurements.

T. asperellum culture filtrates	Inhibition zones (mm)	
Day 3	19.33 ± 0.56ab	
Day 5	25.83 ± 1.09c	
Day 7	30.33 ± 0.19d	
Day 9	22.50 ± 1.31ab	
Day 11	17.83 ± 1.48b	
Day 14	21.00 ± 0.47a	
Ceftazidime (30 µg/mL)	37.67 ± 0.19	
Notes.

The data are represented as the mean ± standard error (SE) from six replicates.

Different superscript letters a, b, c, d indicate statistically significant differences as determined by Tukey–Kramer test at p < 0.05.

Minimum inhibitory concentrations (MICs) and minimum bactericidal concentrations (MBCs) of T. asperellum culture filtrate against B. pseudomallei

The MIC and MBC values of TD7 against B. pseudomallei K96243 were both 7.81 ± 0.00 mg/mL. For B. pseudomallei H777, the MIC was 7.81 ± 0.00 mg/mL and the MBC was 11.72 ± 1.75 mg/mL.

Bactericidal activity of T. asperellum culture filtrate measured by time-kill assay

The relationship between culture filtrate concentration and antibacterial activity could be assessed over time using a time-kill assay. Treatment with 62.48 mg/mL (8 ×MIC) of TD7 demonstrated a potent bactericidal effect, completely eradicating B. pseudomallei K96243 within 30 min (Fig. 1). Reducing the concentration to 31.24 mg/mL (4 ×MIC) and 15.62 mg/mL (2 ×MIC) led to a slower bactericidal effect, with complete loss of bacterial viability occurring within 2 h (Fig. 1). TD7 maintained its bactericidal activity against B. pseudomallei K96243 at a concentration of 7.81 mg/mL (1 ×MIC) but took up to 24 h to kill bacteria (Fig. 1).

Figure 1 Time-kill assay with B. pseudomallei K96243 and varying concentrations of TD7.

Different lines with symbols show results from different TD7 concentrations: untreated control (pink solid line with diamonds), treated with 7.81 mg/mL (green dash-dotted line with squares), 15.62 mg/mL (blue dotted line with circles), 31.24 mg/mL (yellow dashed line with squares) and 62.48 mg/mL (red dashed line with triangles) of TD7. The data are presented as the means ± standard error (SE) from three replicates.

Anti-biofilm activity of T. asperellum culture filtrate against B. pseudomallei biofilm

After treating B. pseudomallei H777 with 3.91 (0.5 ×MIC) and 7.81 mg/mL (1 ×MIC) of TD7 at the adhesion stage (0 h) and biofilm formation stage (24 h), we observed that the higher concentration of TD7 was more effective in controlling biofilm formation compared to the lower concentration at both stages. At a concentration of 7.81 mg/mL, TD7 significantly reduced biofilm formation by 6-fold during the adhesion stage (Fig. 2A) and by 3-fold during the biofilm formation stage (Fig. 2B) compared to the control (one-way ANOVA at adhesion stage: F2,69 = 801.7, p < 2e−16, at biofilm formation stage: F2,69 = 578.9, p < 2e−16).

Figure 2 Quantitative analysis of B. pseudomallei H777 biofilm formation following treatment with TD7 during the adhesion stage (A) and biofilm formation stage (B).

Each treatment included 24 replicates, comprising eight technical replicates from three independent experiments. Each point in the scatter plot represents an individual replicate, while horizontal lines indicate the mean, and vertical lines represent the standard error (SE). Statistically significant differences determined using the Tukey–Kramer test at p < 0.05, were denoted by different letters (a, b).

T. asperellum culture filtrate-induced morphological changes in B. pseudomallei

Scanning electron microscopy of untreated control cells of B. pseudomallei K96243 showed that they had smooth surfaces and intact membranes (Fig. 3A). In contrast, some B. pseudomallei K96243 cells treated with T. asperellum culture filtrate at 3.91 mg/mL (0.5 ×MIC) exhibited a shrunken appearance and a roughened surface (Fig. 3B). The higher the concentration of T. asperellum culture filtrate at 7.81 mg/mL (1 ×MIC), the greater the number of cells exhibiting this shrunken appearance and roughened surface (Fig. 3C).

Figure 3 Scanning electron micrographs (15,000x) of untreated B. pseudomallei K96243 (A), B. pseudomallei K96243 treated with 3.91 mg/mL (0.5xMIC) (B) and with 7.81 mg/mL (1xMIC) TD7 (C).

White arrowheads point to the locations of damaged cells.

T. asperellum culture filtrate-induced alterations in B. pseudomallei membrane permeability

Membrane permeability in treated and untreated cells of B. pseudomallei K96243 was evaluated by the crystal violet uptake assay (Fig. 4). In untreated (negative control) cells, the uptake of crystal violet was 14%. Following treatment with 3.91 mg/mL (0.5 ×MIC) and 7.81 mg/mL (1 ×MIC) of TD7, the uptake increased to 18% and 27%, respectively. In comparison, the uptake of crystal violet in EDTA-treated (positive control) bacteria was 32% (Fig. 4). We found that the crystal violet uptake for the bacteria treated with 3.91 mg/mL and 7.81 mg/mL of TD7 was significantly different from the untreated control (one-way ANOVA: F3,8 = 45.24, p = 2.31e−05).

Figure 4 Crystal violet uptake (%) by B. pseudomallei K96243 cells treated with different concentrations of TD7.

Two different concentrations of TD7 (3.91 mg/mL, blue bar, 7.81 mg/mL, green bar) were tested. Untreated cells (pink bar) were used as a negative control and 0.25M EDTA-treated cells (yellow bar) were used as a positive control. The data are presented as the means ± standard error (SE) from at least four replicates. Different letters (a, b, c) indicate statistically significant differences by Tukey–Kramer test at p < 0.05.

Discussion

Trichoderma species are fungi that are widely used in biocontrol due to their ability to promote plant growth (Wu et al., 2017; Herrera-Téllez et al., 2019), and inhibit microbial growth via mycoparasitism, antibiotic production, and competition for nutrients (Benítez et al., 2004; Vizcaino et al., 2005; Zeilinger et al., 2016; Zhang et al., 2021). This study highlights the antibacterial and anti-biofilm activity of T. asperellum culture filtrates against B. pseudomallei, the etiological agent of melioidosis. As rice farmers are typically at high risk of B. pseudomallei exposure (Limmathurotsakul et al., 2013), and both T. asperellum and B. pseudomallei are microorganisms naturally found in soil, T. asperellum might hold potential as a control agent against B. pseudomallei in endemic areas such as northeastern Thailand. This fungal strain benefits plants and combats phytopathogenic fungi; therefore, farmers may be particularly motivated to adopt its use. By integrating T. asperellum into their agricultural practices, the risk of B. pseudomallei infection could potentially be reduced. Consequently, this practice may contribute to broader public health benefits in regions prone to melioidosis.

We found that T. asperellum culture filtrates collected at different time points exhibit different levels of inhibitory activity against B. pseudomallei, with the filtrate obtained on day 7 (TD7) being the most potent. Our results are consistent with previous studies demonstrating that Trichoderma species exhibit the highest antibiotic production during the log phase of growth, typically between days 3 and 7 (Benítez et al., 2004; Vinale et al., 2014; Zeilinger et al., 2016).

The time-kill assay was used to determine whether T. asperellum culture filtrate components have a bacteriostatic or bactericidal effect against B. pseudomallei. Our results indicate that TD7 exhibits bactericidal activity against B. pseudomallei in both a time- and concentration-dependent manner, i.e., within 30 min at a concentration of 62.48 mg/mL (8 ×MIC), within 2 h at a concentration of 31.24 mg/mL (4 ×MIC) and 15.62 mg/mL (2 ×MIC), and within 24 h at a concentration of 7.81 mg/mL (1 ×MIC). This finding is consistent with previous studies indicating that metabolites with antimicrobial activity, secreted within culture filtrates, effectively kill pathogens in both a time- and concentration-dependent manner (Sangdee et al., 2016; Boottanun et al., 2017; Thammawat, Sangdee & Sangdee, 2017).

Biofilm formation is a significant factor contributing to the antibiotic resistance of B. pseudomallei and this presents a challenge for effective treatment (Sawasdidoln et al., 2010; Sirijant, Sermswan & Wongratanacheewin, 2016). We found that TD7 significantly reduces the ability of B. pseudomallei to establish biofilms during both the adhesion stage (0 h) and the biofilm formation stage (24 h). This effect may be attributable to TD7 inhibiting or killing bacteria before they can adhere to surfaces during the adhesion stage, thereby preventing subsequent biofilm development (Stoodley et al., 2002; Chung & Toh, 2014; Pakkulnan et al., 2019; Pakkulnan, Thonglao & Chareonsudjai, 2023; Shobha et al., 2023; Li et al., 2024). Given that TD7 also inhibits biofilm formation at the 24-hour time point, it may be that TD7 contains substances that disrupt the formation of early biofilm architecture. Disruption of biofilm architecture during the maturation process is an important strategy for eradicating biofilms (Stoodley et al., 2002). Our observation is in accordance with the study of Papaianni et al. (2020) which reported that Trichoderma extracts reduced biofilm formation by the plant pathogenic bacterium Xanthomonas campestris. Their research identified 6-pentyl-α-pyrone (6PP), a metabolite commonly found in various Trichoderma species (Khan et al., 2020), as the key component of an extract that could dissolve biofilms and inhibit their development and maturation. Further research on T. asperellum culture filtrate should be conducted to identify the active components effective against B. pseudomallei and its biofilm.

The morphological alterations of B. pseudomallei observed through SEM in our study suggests that TD7 may affect the bacterial cell wall and cell membrane. The membrane surface irregularities observed in this study closely resemble the alterations in B. pseudomallei treated with metabolites produced by Bacillus amyloliquefaciens KKU1 (Potisap et al., 2018). In the study of B. amyloliquefaciens, the cell wall abnormalities of B. pseudomallei were attributed to protein substances with antibacterial properties, specifically bacteriocins. Bacteriocins possess hydrophobic or amphiphilic properties, targeting bacterial membranes and causing pore formation or complete disintegration of the cell wall (Baker, 1968; Potisap et al., 2018; Wu et al., 2017). The effect of TD7 on membrane permeability was evidenced by the higher uptake of crystal violet dye compared to untreated control cells. Therefore, T. asperellum culture filtrate might contain some lipophilic/hydrophobic substances that preferentially partition from an aqueous phase into bacterial membrane structures, resulting in membrane disruption, increased membrane fluidity and increased permeability (Devi et al., 2010; Khan et al., 2017).

Conclusions

Our findings show that T. asperellum culture filtrates have bactericidal activity and can reduce biofilm formation. This suggests that T. asperellum could be used to reduce the presence of B. pseudomallei in the environment and, consequently, lower the incidence of melioidosis. Thus, T. asperellum is one of potential biological control agents against B. pseudomallei in soil and water environment besides Bacillus amyloliquefaciens KKU1 (Potisap et al., 2018). In recognition that soil is a complex mixture containing diverse microbial communities, it is important to test the efficiency and impact of these biological control agents in real environment. Future studies by our group will thus assess the inhibitory effects of T. asperellum in co-culture with B. pseudomallei in soil. This will aid in evaluating the potential of T. asperellum as a biological control agent against B. pseudomallei in the natural environment.

Supplemental Information

Supplemental Information 1 Antibacterial activity of T. asperellum culture filtrates collected at different time points against B. pseudomallei K96243 assessed by zone of inhibition measurements

The data are represented as the mean ± standard error (SE) from six replicates. Different superscript letters indicate statistically significant differences as determined by Tukey–Kramer test at p < 0.05.

Supplemental Information 2 Time-kill assay with B. pseudomallei K96243 and varying concentrations of TD7

Different lines with symbols show results from different TD7 concentrations: untreated control (pink solid line with diamonds), treated with 7.81 mg/mL (green dash-dotted line with squares), 15.62 mg/mL (blue dotted line with circles), 31.24 mg/mL (yellow dashed line with squares) and 62.48 mg/mL (red dashed line with triangles) of TD7. The data are presented as the means ± standard error (SE) from three replicates.

Supplemental Information 3 Quantitative analysis of B. pseudomallei H777 biofilm formation following treatment with TD7 during the adhesion stage (A) and biofilm formation stage (B)

Each treatment included 24 replicates, comprising eight technical replicates from three independent experiments. Each point in the scatter plot represents an individual replicate, while horizontal lines indicate the mean, and vertical lines represent the standard error (SE). Statistically significant differences were determined using the Tukey–Kramer test at p < 0.05, denoted by different letters.

Supplemental Information 4 Minimum inhibitory concentrations (MICs) and minimum bactericidal concentrations (MBCs) of T. asperellum culture filtrate against B. pseudomallei.

The data are presented as the means ± standard error (SE) from six replicates.

Supplemental Information 5 Crystal violet uptake (%) by B. pseudomallei K96243 cells treated with different concentrations of TD7

Two different concentrations of TD7 (3.91 mg/mL - blue bar and 7.81 mg/mL - green bar) were tested. Untreated cells (pink bar) were used as a negative control and 0.25M EDTA-treated cells (yellow bar) were used as a positive control. The data are presented as the means ± standard error (SE) from at least four replicates. Different letters indicate statistically significant differences by Tukey–Kramer test at p < 0.05.

We would like to thank the Melioidosis Research Center for providing Burkholderia pseudomallei strains K96243 and H777. We also extend our gratitude to the Khon Kaen Agricultural Technology Promotion Center (Plant Protection), Khon Kaen Province, for providing Trichoderma asperellum.

Additional Information and Declarations

Competing Interests

Author Contributions

Data Availability

The authors declare there are no competing interests.

Naritsara Roopkhan conceived and designed the experiments, performed the experiments, analyzed the data, prepared figures and/or tables, authored or reviewed drafts of the article, and approved the final draft.

Thotsapol Chaianunporn conceived and designed the experiments, analyzed the data, authored or reviewed drafts of the article, and approved the final draft.

Sorujsiri Chareonsudjai conceived and designed the experiments, authored or reviewed drafts of the article, and approved the final draft.

Kanokporn Chaianunporn conceived and designed the experiments, performed the experiments, analyzed the data, prepared figures and/or tables, authored or reviewed drafts of the article, and approved the final draft.

The following information was supplied regarding data availability:

The raw data sets used for analysis, providing detailed information on the results of the study, are available in the Supplementary Files.

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
