# Peer review of "Inhibitory effects of Trichoderma asperellum culture filtrates on pathogenic bacteria, Burkholderia pseudomallei"

_PeerJ, doi:10.7717/peerj.19051_

## Round 0.1 · original submission · Major Revisions

Dear Authors,

Thank you for submitting your critical work with us and our peers suggested your work needs to undergo substantial revision. Please go through the reviewer's comments carefully and provide us a point-by-point responses.

We are looking forward to seeing your resubmission.

Best wishes,
Dr. Nagendran Tharmalingam.

Reviewer 1 ·

Basic reporting

This study gives important information and insight on the potential use of Trichoderma asperellum to inhibit Burkholderia pseudomallei.

Experimental design

The aim of the study is clear and appropriate methodology to support the hypothesis. However, the authors may need to give more information on the methods so that the experiment could be reproducible. The discussion and conclusion sections requires restructuring to allow better understanding.

Validity of the findings

The conclusion is too lengthy and needs to be specific to the aim of the study.

Annotated reviews are not available for download in order to protect the identity of reviewers who chose to remain anonymous.

·

Basic reporting

With regard to "Basic reporting" I submit the following recommended edits and comments:

INTRODUCTION
1) Line 53 – 54:
The inclusion of a reference to Seng et al. 2024, does not strictly support the authors claim (stated below):
“The majority of melioidosis patients are rice farmers who have been exposed to B. pseudomallei present in wet soil”.
The referenced paper is of high quality and does refer to a link between agricultural practices and potential exposure of farmers to B. pseudomallei. However, it makes no claim that rice farmers are explicitly the “majority” of patients.

The authors claim here may instead be better supported by referencing a prior article by Limmathurotsakul, where a matched case-control study was performed:
• Limmathurotsakul D, Kanoksil M, Wuthiekanun V, Kitphati R, deStavola B, Day NPJ, et al. (2013) Activities of Daily Living Associated with Acquisition of Melioidosis in Northeast Thailand: A Matched Case-Control Study. PLoS Negl Trop Dis 7(2): e2072. https://doi.org/10.1371/journal.pntd.0002072


2) Line 61 – 63:
The references of Assefa and Amare (2022), and Metzger et al (2022) that follow Line 61 – 63 do not support the preceding sentence (stated below):
“Biofilm formation renders B. pseudomallei less susceptible to many chemical agents, including antibiotics such as doxycycline, ceftazidime, imipenem, and trimethoprim/sulfamethoxazole”

Metzger et al (2022), is concerned with multidrug resistance plasmid transmissibility within biofilms of Acinetobacter baumannii. Assefa and Amare (2022), is concerned with Biofilm-Associated Multi-Drug resistance in hospital-acquired infections. Neither paper mentions B. pseudomallei.

Instead, I would like to suggest that the authors reference the following article by Sawasdidoln et al. (2010), where antibiotic MICs were performed against B. pseudomallei biofilms and planktonic cells.
• Sawasdidoln C, Taweechaisupapong S, Sermswan RW, Tattawasart U, Tungpradabkul S, Wongratanacheewin S (2010) Growing Burkholderia pseudomallei in Biofilm Stimulating Conditions Significantly Induces Antimicrobial Resistance. PLoS ONE 5(2): e9196. https://doi.org/10.1371/journal.pone.0009196


3) Line 71 – 72:
The statement that Trichoderma asperellum exhibits “competitive competence against phytopathogenic fungi” is ambiguous.
The term “competence” may be confusing for microbiologists, as it is typically used to describe the capacity for a microorganism to acquire nucleic acids from the environment.
I suggest the authors use a different term here. The reference by Zeilinger et al (2016) uses the phrase “antagonistic activity”, which may be a suitable substitute.


4) Line 127:
This statement is grammatically incorrect. Please rephrase to:
“T. asperellum culture filtrate was tested in six replicates”


5) Line 280 – 283:
“T. asperellum is already mass-produced and distributed to farmers across northeastern Thailand”

This is important background information for the study and should be included in the introduction.
Are researchers currently monitoring the treatment of agricultural land with T. asperellum?
Are there differences in B. pseudomallei abundance in regions treated with T. asperellum?
Are other microorganisms currently changing in abundance following the introduction of T. asperellum?

6) Supplement data 1: Agar diffusion
Please edit the text of the table to replace “30 ug CAZ” with the correct symbol for “30 microgram CAZ”

7) Supplement data 4: Biofilm test
This table contains the first instance of an “X5 dilution factor”. What is this dilution factor?
I assume that after you solubilise your adherent crystal violet stain in acetic acid, you transfer the solution to a new 96-well plate, and then dilute it 5-fold in additional acetic acid.
If so, please state this in the methods section for the benefit of your readers.

Experimental design

With regard to "Experimental design" I submit the following recommended edits and comments:

1) Line 86 – 87:
“The two bacterial strains, Burkholderia pseudomallei K96243 and B. pseudomallei H777 (a biofilm-producing strain containing the bpsl0618 gene). . .”

More clarity on the strain selection here is needed. The authors phrasing here implies that B. pseudomallei K96243 both does not produce biofilm and does not contain bpsl0618.
However, this is not the case. The strain B. pseudomallei K96243 is a well-studied producer of biofilm and furthermore, bpsl0618 is a K96243 specific locus tag name for a sugar transferase gene present in B. pseudomallei.

Burkholderia pseudomallei strain H777 is a clinical isolate recovered from a blood sample in 2001, and first investigated by Taweechaisupapong et al. (2005). In this study, H777 was classified as a moderate biofilm producer and used to generate biofilm deficient transposon mutants. The mutant strain B. pseudomallei M10, was created by transposon insertion into a sugar transferase within H777 that is homologous to bpsl0618. The creation and history of these strains is discussed by Khamwong et al., (2022).

I recommend that:
1) The original source of H777 is referenced appropriately
2) The reference to bpsl0618 is unnecessary and can be removed. It only makes sense when this strain is compared to a mutant that lacks this gene.

• Khamwong M, Phanthanawiboon S, Salao K, Chareonsudjai S. Burkholderia pseudomallei biofilm phenotypes confined but surviving in neutrophil extracellular traps of varying appearance. Front Immunol. 2022 Aug 18;13:926788. Doi: 10.3389/fimmu.2022.926788. PMID: 36059509; PMCID: PMC9434113.

• Taweechaisupapong, S., Kaewpa, C., Arunyanart, C., Kanla, P., Homchampa, P., Sirisinha, S., Proungvitaya, T., & Wongratanacheewin, S. (2005). Virulence of Burkholderia pseudomallei does not correlate with biofilm formation. Microbial Pathogenesis, 39(3), 77-85. https://doi.org/https://doi.org/10.1016/j.micpath.2005.06.001


2) Line 136 – 149:
“Anti-biofilm activity of T. asperellum culture filtrate against B. pseudomallei biofilm”

Overall, I commend the authors on their detailed protocol. However, I encourage them to include additional information about the following:
• What volume of distilled water was used to wash the biofilm?
• What solvent was used to prepare the 2% w/v crystal violet solution? Or was it purchased pre-prepared?
I refer the authors to a paper produced by Allkja et al., (2020) which provides additional excellent advice on reporting biofilm protocols.

• Allkja, J., Bjarnsholt, T., Coenye, T., Cos, P., Fallarero, A., Harrison, J. J., Lopes, S. P., Oliver, A., Pereira, M. O., Ramage, G., Shirtliff, M. E., Stoodley, P., Webb, J. S., Zaat, S. A. J., Goeres, D. M., & Azevedo, N. F. (2020). Minimum information guideline for spectrophotometric and fluorometric methods to assess biofilm formation in microplates. Biofilm, 2, 100010. https://doi.org/https://doi.org/10.1016/j.bioflm.2019.100010


3) Line 183 – 189:
Antibacterial activity of T. asperellum culture filtrates against B. pseudomallei
Table 1.

The results paragraph does not make clear which strain of B. pseudomallei the authors are assessing. In other results paragraphs, it is very clear whether K96243 or H777 are being assessed. Please include the strain being assessed in both this paragraph, and the accompanying Table.

4) Line 205 – 212:
Anti-biofilm activity of T. asperellum culture filtrate against B. pseudomallei biofilm
Figure 2.

I commend the authors for presenting the raw, unadjusted data for the readers. However, I encourage them to present the data as a scatter graph rather than a bar graph. This would allow for readers to observe the biofilm formation from each of the three independent experiments.

5) Line 214 – 218:
T. asperellum culture filtrate-induced morphological changes in B. pseudomallei
Figure 3.

Please include the details of the strain being assessed in both this paragraph, and the accompanying Figure legend (Line 507 – 509).

Validity of the findings

With regard to "Validity of the findings" I submit the following recommended edits and comments:


1) Line 256 – 258:
“The anti-biofilm mechanism of TD7 against B. pseudomallei could be similar to that of endophytic Trichoderma sp. and Lasiodplodia pseudotheobromae culture filtrates on biofilm-producing MRSA”

The authors should expand on why they believe that the anti-biofilm mechanism of TD7 could be similar to the mentioned examples. The biofilm formation processes, exopolysaccharide compositions, and underlying bacterial physiology of B. pseudomallei and MRSA are very different.


2) Line 259 – 261:
“In these studies, fungal culture filtrates reduced biofilm formation by eliminating bacterial cells before they adhered to surfaces or by altering the surfaces to inhibit bacterial growth, adhesion, and colonization.”

In the references cited, neither Jalil and Ibrahim (2021), or Li et al. (2024), demonstrably prove that the fungal culture filtrates are altering the surface of their environments. There is no mechanistic explanation in these papers as to how the secondary metabolites could be physically or chemically altering the 96-well microtiter plates or coverslips used for biofilm formation. Please rephrase this statement to indicate that this hypothesis has not been conclusively proven.

Additional comments

In summary, with minor exceptions I found this article to be well-written and scientifically robust. I commend the authors for their transparency in providing all raw data. I look forward to the publication of the final article, and future research expanding on the complex interactions between B. pseudomallei and other microorganisms within the soil environment.

Reviewer 3 ·

Basic reporting

Overall, the manuscript was written in a clear and easy to understand English. Some simple grammatical error which can be edited using grammar tools. The literature is well referenced with the background on both Burkholderia pseudomallei and Trichoderma asperellum provided. The manuscript is written with standard structure and all figures, tables and respectively legends labelled.
The manuscript is well written to describe the effects of the Trichoderma asperellum extracts against B.pseudomallei.

Experimental design

The methods described for the study is detailed, covering all aspects needed to achieve the objective of the study. However, here are some minor issues highlighted
1. Please reassess the standard unit i.e : ml should be mL and other throughout the manuscript.
2.The method for preparation for scanning electron microscope should be mentioned and referenced.
3. Please briefly explain the terminology 1xMIC, 2X MIC

Validity of the findings

1.The results provided are valid and correlate with the tables and figures given.
2.The data have been statistically validated.
3. The conclusion section need to be edited extensively as it seems more like discussion.
4.The study limitations and future research from this study should be included in the discussion section.
5. The authors should discuss the use of the T. asperellum in agriculture practices and its impact against plant pathogens- followed by the impact of B. pseudomallei infection among farmers and then correlate with the potention impact of the T. aspellum use.
6. I dont see any discussion on the SEM work performed.

---

## Round 0.2 · Minor Revisions

Dear Authors,

Thank you for submitting the revised version of your manuscript. We appreciate your efforts and the improvements made so far.

Reviewer 1 has suggested a few minor changes that need to be addressed. Kindly review these comments and revise your manuscript accordingly. Please include a detailed, point-by-point response to the reviewer’s suggestions when submitting your revised version.

We look forward to receiving your updated manuscript soon.

Best regards,
Dr. Nagendran Tharmalingam
Handling Editor

Reviewer 1 ·

Basic reporting

Overall, the authors did a significant improvement on the whole manuscript. The current version is clear and structured well. I commend on the authors' effort on this. Good job! I have only a few comments for the authors.

Experimental design

The aim of the manuscript is addressed and supported by the results.

Validity of the findings

The methodology is clear and allow reproducibility.

Additional comments

I have a few comments to the authors.

Line 63: Biofilm formation renders B. pseudomallei less susceptible to many
chemical agents, including antibiotics such as doxycycline, ceftazidime, imipenem, and trimethoprim/ sulfamethoxazole. This study focus more on the environment. Therefore, I suggest the authors to include examples on the effect of biofilm formation to chemicals used in soil or water apart from clinical setting.

Line 70: Please define Trichoderma asperellum e.g. fungi.
Line 75: The reference Khamkha, 2019 is not accessible. Please give alternative, if any. This information is important to emphasize the current use of T. asperellum as biological agent.

Line 120: Are the culture filtrates prepared fresh? Please also include the respective days of culture.

Line 141: I suggest the authors to remove the wording in the column to avoid confusion. e.g. double the MIC. Instead, state that different concentrations were used.

Line 152: What is the starting concentration of B. pseudomallei and T. asperellum used?

Line 161: Is this statement correct? " Excess stain was gently washed with running tap water until clear". The authors were using 96 well plate for this experiment.

Line 166: Please give information on this statement. "The values were then calculated to obtain the original optical density."

Line 234: The authors may reconsider to use the term "dose-dependent manner" as only two concentrations were tested.

Line 244: The authors may directly specify the concentrations used as only two concentrations were tested.

Line 262: The authors may change to this: This study highlights the antibacterial and anti-biofilm activity of T. asperellum culture filtrates against B. pseudomallei, the etiological agent of melioidosis.

Line 266-270: These sentences need to be rephrased.

Line 279: Please include also results for 2X and 4X MIC to show the decreasing trend.

Line 303: The authors may consider to use a different word instead of "membrane roughness."

Line 315: Is there studies on co-culture of other organism with BP showing potential antibacterial activity in soil or water environment? The authors may include in discussion. Limitations may be included in the last paragraph of discussion.

Figures and tables:
The authors need to define what different superscript letters referring to in the footnote.

·

Basic reporting

I am satisfied with the edits the authors have made regarding basic reporting.

Experimental design

I am satisfied with the edits the authors have made regarding experimental design.

Validity of the findings

I am satisfied with the edits the authors have made regarding the validity of their findings.

Additional comments

Thank you for your diligent efforts to edit this manuscript.

---

## Round 0.3 · accepted · Accept

Dear Authors,

I am writing to inform you that your manuscript has been accepted for publication. Congratulations!

The production house will contact you for typeset-related queries. We are looking forward to seeing your future submissions.

Best wishes.
Dr. Nagendran Tharmalingam.